# Skeletal Changes in Growing Cleft Patients with Class III Malocclusion Treated with Bone Anchored Maxillary Protraction—A 3.5-Year Follow-Up

**DOI:** 10.3390/jcm10040750

**Published:** 2021-02-13

**Authors:** Ralph M. Steegman, Annemarlien Faye Klein Meulekamp, Arjan Dieters, Johan Jansma, Wicher J. van der Meer, Yijin Ren

**Affiliations:** 1Department of Orthodontics, University of Groningen, University Medical Center Groningen, 9713 GZ Groningen, The Netherlands; r.m.steegman@umcg.nl (R.M.S.); a.f.klein.meulekamp@umcg.nl (A.F.K.M.); j.a.dieters@umcg.nl (A.D.); w.j.van.der.meer@umcg.nl (W.J.v.d.M.); 2Department of Oral Maxillofacial Surgery, University of Groningen, University Medical Center Groningen, 9713 GZ Groningen, The Netherlands; j.jansma@umcg.nl; 3Department of Orthodontics, W.J. Kolff Institute, University of Groningen, University Medical Center Groningen, 9713 GZ Groningen, The Netherlands

**Keywords:** bone anchored, maxillofacial protraction, color mapping, 3D superimposition, cleft, orthodontics, class III malocclusion, CBCT, orthopedic therapy

## Abstract

This prospective controlled trial aimed to evaluate the skeletal effect of 3.5-years bone anchored maxillary protraction (BAMP) in growing cleft subjects with a Class III malocclusion. Subjects and Method: Nineteen cleft patients (11.4 ± 0.7-years) were included from whom cone beam computed tomography (CBCT) scans were taken before the start of BAMP (T0), 1.5-years after (T1) and 3.5 y after (T2). Seventeen age- and malocclusion-matched, untreated cleft subjects with cephalograms available at T0 and T2 served as the control group. Three dimensional skeletal changes were measured qualitatively and quantitatively on CBCT scans. Two dimensional measurements were made on cephalograms. Results: Significant positive effects have been observed on the zygomaticomaxillary complex. Specifically, the A-point showed a displacement of 2.7 mm ± 0.9 mm from T0 to T2 (*p* < 0.05). A displacement of 3.8 mm ± 1.2 mm was observed in the zygoma regions (*p* < 0.05). On the cephalograms significant differences at T2 were observed between the BAMP and the control subjects in Wits, gonial angle, and overjet (*p* < 0.05), all in favor of the treatment of Class III malocclusion. The changes taking place in the two consecutive periods (ΔT1-T0, ΔT2-T1) did not differ, indicating that not only were the positive results from the first 1.5-years maintained, but continuous orthopedic effects were also achieved in the following 2-years. Conclusions: In conclusion, findings from the present prospective study with a 3.5-years follow-up provide the first evidence to support BAMP as an effective and reliable treatment option for growing cleft subjects with mild to moderate Class III malocclusion up to 15-years old.

## 1. Introduction

Children born with cleft lip and palate (CLP) are usually characterized by a Class III malocclusion, which is primarily the result of maxillary deficiency. Maxillary deficiency in growing subjects, when left untreated, can lead to severe functional, aesthetic, breathing and psychological problems [1,2,3,4,5,6]. Important treatment goals in cleft patients are to establish a normal dental occlusion and to correct the skeletal discrepancy for psychological and functional well-being [7,8,9,10].

Currently, the most common orthopedic treatment in both cleft and non-cleft patients with a Class III malocclusion is facemask therapy (FM) with or without rapid maxillary expansion (RME). This tooth-borne treatment modality, often used in patients with early deciduous or early mixed dentition [11,12], has been reported to have undesirable side effects such as dentoalveolar compensation and a clockwise rotation of the mandible [13,14]. These side effects are caused by the induced orthopedic force, which is not transferred directly to the circum-maxillary sutures, but directly or partially on the dentition.

Bone anchored maxillary protraction (BAMP) is an increasingly used treatment modality in cleft patients with Class III malocclusion and can be started in the late mixed- or early permanent-dentition [15]. Previous studies reported positive outcomes on 1.5-years BAMP-treatment in growing CLP subjects with Class III malocclusion. Significant orthopedic changes were observed in the maxillary and zygomatic areas, in addition to the favorable dentofacial aesthetic changes [16,17]. A unique advantage of BAMP, compared to other conventional treatment modalities for Class III malocclusion, is that it transfers forces from bony structures directly onto the sutural sites of the zygomaticomaxillary complex. With this, the treatment effect achieved is mainly skeletal orthopedic with minimal undesirable dental compensation [18]. Additional advantages of BAMP over FM are the continuous light forces used rather than heavy intermittent forces and the lower degree of burden for the patient in terms of compliance. However, it remains to be seen whether the positive results of the first 1.5-years will persist, and whether more orthopedic gains can be achieved when patients reach the end of their growth spurt.

As BAMP treatment is a relatively new technique, little is known about its long-term clinical effects. To date, no results of BAMP treatment in cleft subjects longer than 1.5 y have been reported in literature. The aim of this prospective controlled trial is to investigate the 3D skeletal effect after 3.5 y of BAMP treatment in growing cleft patients using cone beam computed tomography (CBCT) scans.

## 2. Subjects and Methods

### 2.1. Trial Registration and Ethical Approval

This prospective controlled trial is registered at The Netherlands National Trial Registration (TC 6559) and at the Clinical Study Register of the University Medical Center Groningen (201700423). Ethical approval has been granted by the Ethics Committee at the UMCG (METc 2018/318).

### 2.2. Treatment and Control Subjects

A power analysis on the minimal number of subjects needed to detect a difference between the BAMP and the control group at T2 (3.5 y), with a power of 80% and *p* < 0.05, showed that 11 participants were needed to detect a difference on A-point Region of Interest (ROI) from CBCT surface models, and 17 participants to detect a difference on ANB angle from cephalograms. Therefore, the minimal number of subjects per group was set at 17 for the present study. Twenty-three consecutively treated cleft patients were included in the study. All 23 patients have undergone a series of interdisciplinary treatments at the University Medical Center Groningen (UMCG) in the Netherlands under the same set of protocols. Orthodontic treatment was performed by the same orthodontist (Y.R.) for all subjects at the Department of Orthodontics. Bollard bone plates were inserted by the same oral surgeon (J.J.) at the Department of Oral and Maxillofacial Surgery. The inclusion criteria were the same as in our previous study [16], namely: (1) either a complete unilateral cleft lip, palate, or both; (2) either a sagittal overjet between +2 and −5 mm, an ANB angle <0°, or both, or a Wits <0 mm; (3) a secondary bone transplantation prior to BAMP; (4) no or light dental alignment in the upper arch in preparation for bone transplantation, prior to BAMP; (5) both lower permanent canines have been erupted; and (6) no forced overclosure or functional shift present. From all study subjects an informed consent was obtained.

Due to ethical reasons, no CBCT scans from untreated cleft subjects could be obtained. Instead, 2D cephalograms were included from an age- and malocclusion-matched cleft lip and palate group at T0 and T2. This control group was derived from the clinical database of the Department of Orthodontics of the UMCG. All subjects signed an informed consent for the use of their X-ray photos and clinical photos for research purposes. Inclusion criteria for the control group were the same as the BAMP group except that the eruption status of the lower permanent canines was not used as a criterion.

Of the 23 initially included cleft lip and palate patients, four had to be excluded from data analysis due to CBCT acquisition errors. The age of the 19 included study subjects was 11.4 ± 0.7 y at T0. For the control group, a total of 17 patients were included who met the inclusion criteria. The mean age of the control subjects was 10.7 ± 1.2 y at T0. With regard to the protocol of wearing the elastics, a high degree of compliance was recorded for all subjects based on self-report. In 7 patients a removable bite plate was used to temporarily remove the occlusal interference.

### 2.3. Bone Anchored Maxillary Protraction

According to the published protocol, four Bollard bone plates (Bollard, Tita-Link, Brussels, Belgium) were placed under general anesthesia at the age of about 11 [16,17]. Two Bollard bone plates were placed onto the zygomatic buttresses and two on the anterior part of the mandible between the lateral incisor and canine. Three weeks after the placement, maxillary protraction started with intermaxillary elastics with a force of about 150 g in the first month followed by a force of 200–250 g after 2–3 months (Figure 1). Patients were instructed to wear the elastics for 24 h per day, 7 d a week, including meals, and to replace the elastics once a day.

### 2.4. CBCT Scans

All cone beam computed tomography (CBCT) scans were made by the same experienced X-ray technician (A.D.), before the start of BAMP as part of a standardized documentation for diagnostic purposes (T0), 1.5 y after BAMP (T1) and 3.5 y after BAMP treatment (T2). The exact timing of all scans was aligned as much as possible to the need for a CBCT scan for diagnostic purposes or treatment progress evaluations in order to minimize the total number of radiographs subscribed for each individual subject. All patients were positioned in the CBCT scanner in a sitting position with the Frankfurt horizontal plane (FH) parallel to the floor and positioned centrally in the field of view (FoV) using laser alignment. Subjects were instructed not to move, not to swallow and to breathe normally during the acquisition. The CBCT scans were performed using the Planmeca ProMax 3D set at 90 kV and 20.25 mAs while using a 170 × 200 mm FoV and with an isotropic voxel size of 0.3 mm. Acquisition time was 9.0 s in all scans.

### 2.5. Superimposition of the CBCT 3D Surface Models

The Digital Imaging and Communications in Medicine (DICOM) data resulting from each CBCT scan were exported to specialized software (Mimics 10.01, V10.2.1.2 by Materialise Inc., HQ Leuven. Belgium) for hard tissue segmentation to create a 3D surface polygon model, saved as a stereolithography (STL) file. Three-dimensional surface models from the same subjects were imported into Geomagic (version 2013.0.1.1206, Geomagic Solutions, Rock Hill, CA, USA) for a three-dimensional superimposition and color mapping comparison between T1 and T0 and between T2 and T0. The T0 surface model served as the reference model, whereas the T1 and T2 surface models served as the test models. For superimposition, the anterior cranial base and the occipital area posterior of the foramen magnum were selected as the stable structures (Figure 2A) [19].

The predefined Regions of Interest (ROI) measurements were performed on a color map of the two superimposed surface models, namely nasion (N), the left and right zygomatic processes (Zyg), A-point, the right upper central incisors (U1), B-point, pogonion (Pg) and menton (Me) (Figure 2B). Each ROI consists of an area of 4.5 mm^2^ around a pre-defined anatomical landmark containing approximately 60 polygons. Measurement outcomes are the average of these 60 included points at the respective ROIs. The superimposition was set to be stable if the deviations of the ROI at the cranial base and posterior of the foramen magnum were within the range of −0.3 and +0.3 mm, the same as the voxel size. The overall displacement of an ROI is further divided into: the horizontal component (y-axis), the vertical component (z-axis) and the transverse component (x-axis).

### 2.6. Tracing on Lateral Cephalograms

From all CBCT scans of the BAMP group at T0, T1 and T2, a lateral cephalogram was extracted according to an established protocol [20]. All tracings were performed in ViewBox (version 3.00 dHal software, Demetrios Halazonetis, Kifissia, Greece). Before tracing, all lateral cephalograms were set at the same scale. A standard set of cephalometric measurements was included.

### 2.7. Quantitative Measurements

In the BAMP subjects both 3D and 2D quantitative measurements were performed based on either the CBCT surface models or lateral cephalograms derived from the CBCT scans. All 3D measurements were carried out twice by two observers (R.S. and A.K.M.), with an interval of at least one week to reduce the risk of bias. A maximum of 15 measurements per day were conducted to prevent a possible effect of fatigue. In the control subjects only 2D quantitative measurements on the lateral cephalogram were measured. All 2D measurements were carried out twice by one observer (A.K.M.) at an interval of at least one week. The CBCT surface models or cephalograms of different subjects were randomly assigned to the observers during the measurements.

### 2.8. Statistics

Statistical analyses were performed with SPSS (version 23.0, IBM, Armonk, NY, USA) by one author (R.S.). Cephalometric data were tested with a Kolmogorov–Smirnov test for normal distribution. All cephalometric values were normally distributed at T0. Furthermore, 3D and 2D data were tested with a one-way ANOVA, with a Tukey’s HSD post hoc test for significances. The significance level for all tests was set to *p* < 0.05. For the intra- and interclass correlation a Cronbach’s alpha test was used, with a kappa of 0.81–1.00 indicating a “near perfect agreement”.

## 3. Results

The inter-observer agreement of all 3D measurements (ICC > 0.84) and the intra-observer agreement of all 2D measurements (ICC > 0.90) were both “near perfect”.

### 3.1. D Skeletal Changes

An illustration of skeletal changes after 1.5-years and 3.5-years BAMP treatment on a 3D color map is given in Figure 3. The zygomatic and maxillary anterior regions show more intense yellow, orange and red colors in the entire 3.5-year period (ΔT2-T0) than in the first 1.5-year period (ΔT1-T0), indicating more forward and outward displacements. 

In Figure 4, skeletal changes on superimposed 3D surface models are visualized from a “median patient”, namely, the patient whose displacement of A-point ROI represents the median value among all 19 subjects from the BAMP group. From a frontal and lateral view (Figure 4 A,B,E,F), forward, outward and downward displacements of the zygomaticomaxillary complex could be observed at both 1.5-years and 3.5-years, with the latter more pronounced. Similar observations could be seen from the axial views of the zygoma arches and the mandibular body (Figure 4 C,D,G,H).

Table 1 presents the 3D quantitative data measured on the colormap. On superimposed CBCT surface models, except for the B-point and the upper incisor, all other studied ROIs showed significantly more displacements in the 3.5-year period (ΔT2-T0) than in the first 1.5-year period (ΔT1-T0) (*p* < 0.05). The significant changes at A-point and nasion at 3.5-years (2.7 mm ± 0.9 mm and 1.7 mm ± 1.1 mm, respectively) are mainly attributed to the horizontal component (2.6 mm ± 0.8 mm and 1.6 mm ± 1.1 mm, respectively), indicating forward and outward displacements (*p* < 0.05). Zygoma regions showed significant displacements in all three dimensions (horizontal, vertical and transverse) (*p* < 0.05). B-point is the only region that did not show any significant displacement at either 1.5-years or 3.5-years after BAMP treatment.

Interestingly, comparisons of the changes occurring in the two consecutive observation periods only showed differences in the upper incisor with less displacement in the second 2-year period (ΔT2-T1, 0.5 mm ± 1.6 mm) than in the first 1.5-year period (ΔT1-T0, 2.3 mm ± 1.4 mm) (*p* < 0.05). The zygoma regions also showed more vertical displacement in the second period (ΔT2-T1, 1.0 mm ± 0.6 mm) than in the first period (ΔT1-T0, 0.4 mm ± 0.4 mm) (*p* < 0.05).

### 3.2. 2D Cephalometric Changes

All cephalometric measurements and changes in the BAMP group and the control group are presented in Table 2. In the control group the U1-palatal plane and U1-NA were significantly lower at T0 than at T2 (96.3° ± 6.9° vs. 107.9° ± 8.0°, *p* < 0.001). Consequently a significantly higher inter-incisor angle was found at T0 than at T2 (149.0° ± 9.6° vs. 138.5° ± 12.2°, *p* < 0.05). Interestingly, at T0 there were significant differences between the BAMP and control group on exactly the same three parameters (*p* < 0.05). Nevertheless, all these differences between the two groups at T0 disappeared at T2. Noteworthy, was that the significant differences observed at T2 between the BAMP and control group were in Wits (−0.4 mm ± 4.0 mm vs. −4.3 mm ± 4.7 mm) (*p* < 0.05), gonial angle (126.6° ± 7.0° vs. 133.1° ± 7.3°) (*p* < 0.05) and overjet (2.2 mm ± 2.8 mm vs. −2.6 mm ± 4.1 mm) (*p* < 0.01); all appeared to favor orthopedic treatment of Class III malocclusion. Regarding the net changes between T2 and T0, the control group showed a tendency for mild improvement in overjet and mild deterioration in ANB and Wits, all indicating a temporary dental camouflage. Over the 3.5-year period (ΔT2-T0) the most significant difference between the BAMP and control group was observed in the ANB angle (−2.2 ± 2.3 vs. 1.1 ± 2.1, *p* < 0.000), with an average improvement of 3.3°.

Within the BAMP group, only one single parameter, overjet, showed significant changes between T0 and T2, from −1.3 mm ± 2.7 mm at the start of BAMP to 2.2 mm ± 2.8 mm 3.5-years after (*p* < 0.05), an average increase of 3.5 mm. Changes taking place in the two consecutive periods (ΔT1-T0 and ΔT2-T1) showed no difference in any measurements in the BAMP group.

## 4. Discussion

Application of BAMP treatment modality in cleft subjects is relatively new. A recent systematic review of orthopedic therapies in growing cleft patients, only included five studies using BAMP therapy with treatment durations ranging from 7.9 to 18 months [21]. To our knowledge, the present study is the first in which both 3D and 2D effects of BAMP treatment on the zygomaticomaxillary and mandibular structures were evaluated with a follow-up of 3.5-years.

Our previous study on 1.5-years of follow-up showed a significant difference between the BAMP group and untreated cleft control group in ANB angle, SNA angle, Wits and overjet (*p* < 0.05), all in favor of improvement of Class III malocclusion by BAMP therapy. [16] Here, with a follow-up of 3.5-years, the difference between the BAMP and the control group is most notable at the skeletal level as measured by the ANB angle, with an average increase of 3.3°; this is not only statistically significant but also highly clinically meaningful.

Although the upper incisor inclinations were not the same in the BAMP and control group at T0, these differences disappeared at T2. The more retroclined upper incisor in the control group to start with, became less retroclined after 3.5-years, explaining the mild improvement in negative overjet at T2 in the control group. Although the dental relationship appeared mildly improved in the control subjects, their skeletal Class III relationship had actually worsened after 3.5-years as seen in the decreased ANB and Wits.

Since the changes ΔT2-T0 in the control group can largely be considered the result of growth, the observed difference between the BAMP and control group can therefore be attributed to the effect of BAMP treatment. Had this study not included a control group, this important effect of growth would have been overlooked. That said, it is understandable that comparisons within the BAMP group at different time points were mostly statistically insignificant, as any deterioration due to normal growth would have camouflaged the positive treatment effect. The true treatment effect can only be appreciated by comparing the differences between the BAMP group and the control group over the same observation period. Nonetheless, we observed significant differences at T2 between the BAMP and control group in Wits, gonial angle and overjet, with an average change of +4.7 mm, −6.5° and +4.8 mm, respectively, all in favor of BAMP treatment of Class III malocclusion.

An important finding in the present study is that changes that occured in the two consecutive periods (ΔT1-T0 and ΔT2-T1) did not show a difference in the BAMP group in any measurements, with the exception of the upper inclination of the upper incisor. This means that not only the positive results from the first 1.5-years were maintained, but also continuous orthopedic effect was achieved in the following 2-years. In other words, treatment efficacy of BAMP therapy observed in the first period continues to the same extent in the second period. This is unexpected, as the growth potential decreases with the progress of treatment. By the end of the 3.5-year observation period, the average age of the study subjects was close to 15, typically at the tail of the growth spurt. These results seem to indicate that the optimal age to start BAMP therapy may warrant a critical evaluation. Oppositely, it can also imply that the age of 11 to 12-years is indeed the optimum, as continuous treatment benefit is ensured when the treatment has started by then. More extensive studies are needed to clarify the optimal timing.

Another important finding is that the overall displacement of the upper incisor, ΔT2-T1 is less than ΔT1-T0, indicating that most of the possible dental compensation effect occurred in the first period, as was also observed in the control group, and that any improvement in the position of the upper incisor observed in the second period is mainly due to the skeletal effect.

In the vertical dimension, a significant decrease of 6.5° in the gonial angle was observed in the BAMP group compared to the control. Such a “closing” effect of the gonial angle has been previously reported in non-cleft subjects after 1.5-years of BAMP treatment [22]. Additionally, other vertical angular measurements from the BAMP group, such as ANS-PNS/GoGn and SN/GoGn, showed a tendency for a greater decrease compared to the control group. These results point out that BAMP treatment did not cause clockwise rotation of the mandible, which is a common observation related to the treatment mechanics for Class III malocclusion [23]. BAMP therapy appears to improve the skeletal relationship by shifting the differential growth of the maxilla and mandible to a more convex direction, namely, more relative growth in the maxilla and less in the mandible. As discussed previously, BAMP therapy likely stimulates more forward growth of the maxilla by opening the palatomaxillary suture and zygomaticomaxillary suture [18,24]. However, the underlying mechanism how the relatively less growth of the mandible is realized has not been much investigated. One hypothesis could be the remodeling of the condylar head and the fossa due to the light compression force on the condyles resulting from the intermaxillary elastics, which reduces the gonial angle producing an effect of the mandible moving “backward” and “counter-clockwise”, as supported by the slightly reduced occlusal plane angle (occlusal-SN) in addition to a significantly reduced gonial angle.

Large variations exist in the individual subjects regarding the effect of the treatment. For example, in the BAMP group, a displacement of A-point has been observed with a maximum of 6.5 mm and a minimum of −0.8 mm. Such a large individual variation could be caused by zygomatic suture maturation, with more favorable results reported for patients at an early stage of maturation [25]. According to Angelierri et al., the zygomaticomaxillary sutures show a high level of fusion at the chronological age of 15-years and older. However, fusion of the sutures has also been observed in younger patients [15].

Several limitations of this study need to be acknowledged. First is our choice for the control group. It would be ideal to have CBCT surface models from an untreated cleft group matched by age, gender and skeletal deformity to understand the effect of BAMP therapy at a three-dimensional level. However, ethical reasons made it impossible to include such a group. As an alternative we used two-dimensional cephalograms from a cleft group as the control. One limitation is that we could not visualize the effect of growth in 3D color mapping either during the entire observation period or in the two respective observation periods. Another limitation is the sample size of the study. Although the number of subjects in both the BAMP and the control groups was above the required minimum, they are both relatively small. Given the lack of a control group with 3D data, caution should be taken when drawing conclusions from the 3D results. The favorable changes observed in the BAMP group is a combined result of the normal growth and the orthopedic effect of the BAMP therapy throughout the entire observation period. More independent studies with a larger scope are needed to verify the findings of the current study, and more importantly, to identify predictable factors of individual characteristics for a favorable treatment outcome of BAMP therapy.

## 5. Conclusions

In conclusion, this prospective study with a follow-up of 3.5-years showed significant positive effects on the zygomaticomaxillary complex and in the vertical dimension of the mandible, providing the first evidence to support BAMP as an effective and reliable treatment option for growing cleft subjects with mild to moderate Class III malocclusion up to 15-years of age.

## Figures and Tables

**Figure 1 jcm-10-00750-f001:**
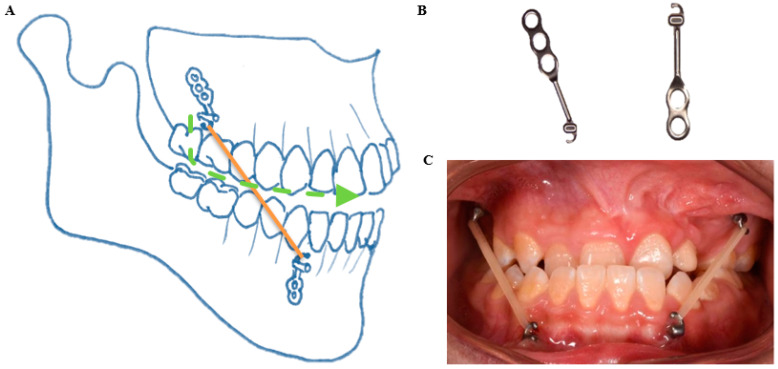
Bollard bone plates used in the present study for bone anchored maxillary protraction. (**A**) A schematic representation of the locations of the Bollard bone plates. (**B**) Bollard bone plates, left for the upper, right for the lower. (**C**) An intra-oral picture of bone anchored maxillary protraction with intermaxillary elastics.

**Figure 2 jcm-10-00750-f002:**
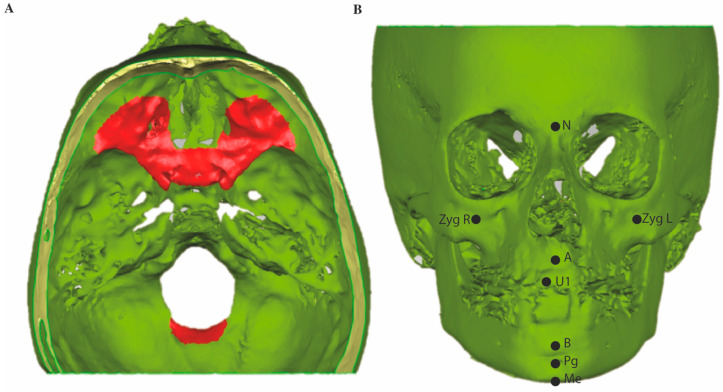
Stable structures used for superimposition and Regions of Interest on cone beam computed tomography (CBCT) 3D surface models. (**A**) Anterior cranial base and posterior of the foramen magnum were selected (areas indicated in red) as the stable structures for superimposition [14]. (**B**) ROIs of A-point (A), B-point (B), nasion (N), menton (Me), pogonion (Pg), right upper incisor (U1), zygoma left and right (Zyg L, Zyg R), respectively.

**Figure 3 jcm-10-00750-f003:**
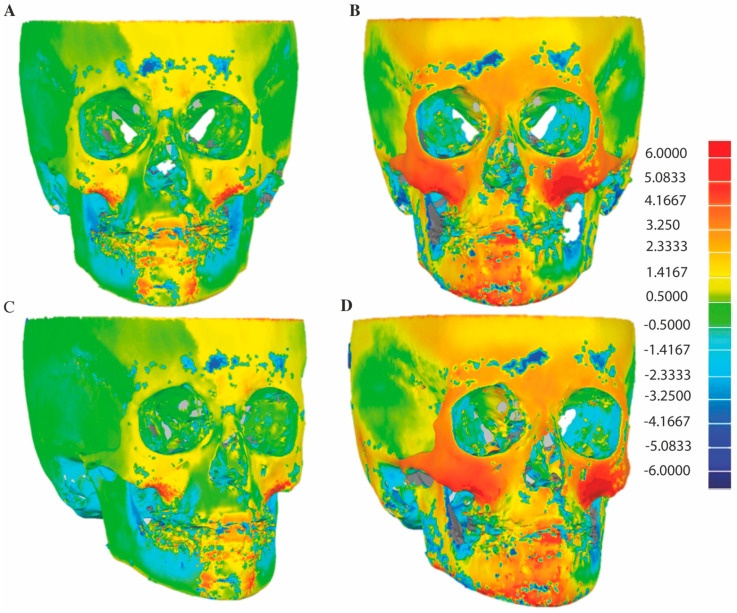
An example of illustration of skeletal changes on 3D colormap. (**A**,**C**) superimpositions of T0 and T1, (**B**,**D**) superimpositions of T0 and T2.

**Figure 4 jcm-10-00750-f004:**
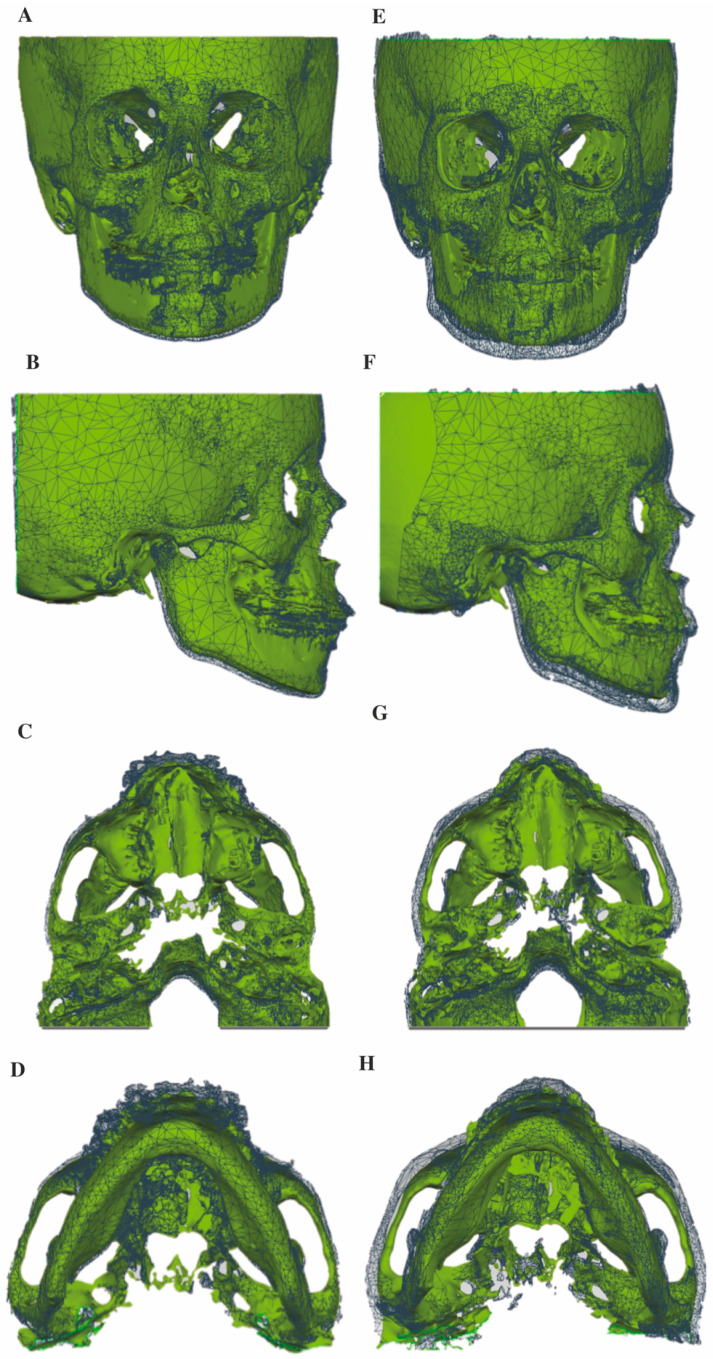
Skeletal changes on superimposed 3D STL surface models from a “median” patient. (**A**–**D**) Superimpositions of T0 and T1, (**E**,**F**) superimpositions of T0 and T2; (**A**,**E**) frontal view, (**B**,**F**) sagittal view, (**C**,**G**) axial view from the cranial base and (**D**,**H**) axial view from the mandible body. T0 STL surface models in green and T1 or T2 in grey mesh.

**Table 1 jcm-10-00750-t001:** Three-dimensional skeletal changes based on CBCT surface models.

N = 19		T1-T0	T2-T1	T2-T0
		1.5-years	2-years	3.5-years
		Mean (SD)	Mean (SD)	Mean (SD)
A	overall	1.5 ± 1.5	0.9 ± 1.7	2.7 ± 0.9 *
	horizontal	1.5 ± 1.4	0.8 ± 1.5	2.6 ± 0.8 *
	vertical	−0.2 ± 0.5	0.1 ± 0.4	−0.1 ± 0.4
B	overall	1.2 ± 1.9	1.1 ± 1.9	2.4 ± 1.2
	horizontal	1.2 ± 1.9	1.0 ± 1.9	2.3 ± 1.2
	vertical	0.1 ± 0.4	0.0 ± 0.4	0.1 ± 0.4
Zygoma left	overall	1.7 ± 0.9	2.2 ± 1.3	3.9 ± 1.3 *
	horizontal	1.3 ± 0.7	1.7 ± 1.0	3.1 ± 1.0 *
	vertical	0.4 ± 0.4	1.0 ± 0.7 #	1.4 ± 0.7 *
	transversal	1.0 ± 0.7	0.8 ± 0.8	1.6 ± 0.9 *
Zygoma right	overall	1.6 ± 1.2	2.0 ± 1.4	3.7 ± 1.1 *
	horizontal	1.2 ± 1.1	1.6 ± 1.3	3.0 ± 1.2 *
	vertical	0.4 ± 0.5	1.1 ± 0.7 #	1.5 ± 0.6 *
	transversal	0.6 ± 0.6	0.6 ± 0.5	1.2 ± 0.4 *
Zygoma overall	overall	1.7 ± 1.0	2.1 ± 1.2	3.8 ± 1.2 *
	horizontal	1.3 ± 0.9	1.6 ± 1.1	3.0 ± 0.9 *
	vertical	0.4 ± 0.4	1.0 ± 0.6 #	1.4 ± 0.7 *
	transversal	0.8 ± 0.6	0.7 ± 0.5	1.4 ± 0.7 *
Nasion	overall	0.7 ± 1.2	1.0 ± 1.1	1.7 ± 1.1 *
	horizontal	0.6 ± 1.1	1.0 ± 1.1	1.6 ± 1.1 *
	vertical	0.0 ± 0.3	0.1 ± 0.5	0.1 ± 0.3
Pogonion	overall	1.0 ± 2.0	1.3 ± 1.5	2.3 ± 1.5 *
	horizontal	1.0 ± 1.9	1.1 ± 1.4	2.0 ± 1.3
	vertical	0.2 ± 0.3	0.5 ± 0.7	0.7 ± 0.6 *
Menton	overall	1.6 ± 1.7	1.4 ± 2.0	3.1 ± 1.5 *
	horizontal	1.0 ± 1.1	0.6 ± 1.2	1.5 ± 0.8
	vertical	1.2 ± 1.4	1.3 ± 1.5	2.5 ± 1.3 *
U1	overall	2.3 ± 1.4	0.5 ± 1.6 #	2.8 ± 1.2
	horizontal	2.1 ± 1.4	0.4 ± 1.5 #	2.5 ± 1.1
	vertical	0.1 ± 0.6	−0.3 ± 0.7	−0.3 ± 1.0

The 3D surface model of T0 was set as the reference model, whereas T1 and T2 were set to be the test models. A difference is considered significant if *p* < 0.05. * indicates significant difference between ΔT2-T0 and ΔT1-T0, # indicates significant difference between ΔT2-T1 and ΔT1-T0.

**Table 2 jcm-10-00750-t002:** Cephalometric measurements of the BAMP treated group and the control group.

	Control Group (N = 17)			Bone Anchored Maxillary Protraction Group (N = 19)					
	T0	T2	ΔT2-T0	T0	T1	T2	ΔT1-T0	ΔT2-T1	ΔT2-T0
Age	10.7 ± 1.2	14.3 ± 1.4	3.7 ± 0.7	11.4 ± 0.7	12.9 ± 0.8	15.0 ± 0.7	1.5 ± 0.4	2.1 ± 0.7	3.6 ± 0.7
SN-FH (°)	10.5 ± 4.0	8.5 ± 5.8	−2.0 ± 5.0	10.4 ± 4.2	10.4 ± 4.2	10.6 ± 3.8	0.0 ± 1.8	0.2 ± 2.7	0.2 ± 2.6
SNA (°)	75.5 ± 5.5	75.2 ± 5.1	−0.3 ± 3.5	77.1 ± 5.3	78.6 ± 5.2	78.6 ± 5.8	1.6 ± 1.5	0.0 ± 2.5	1.6 ± 2.7
SNB (°)	75.9 ± 4.5	77.8 ± 5.7	1.9 ± 3.8	78.1 ± 4.1	78.5 ± 4.1	78.5 ± 4.9	0.4 ± 2.2	0.1 ± 2.8	0.5 ± 2.2
ANB (°)	−0.4 ± 3.5	−2.6 ± 2.7	−2.2 ± 2.3	−1.0 ± 3.1	0.1 ± 3.1	0.1 ± 3.2	1.1 ± 1.7	0.1 ± 1.9	1.1 ± 2.1 @
Wits (mm)	−2.9 ± 2.8	−4.3 ± 4.7	−1.4 ± 4.9	−1.8 ± 3.4	−0.2 ± 3.4	−0.4 ± 4.0 @	1.6 ± 2.2	−0.2 ± 2.9	1.4 ± 2.9
ANS-PNS/Go-Gn (°)	27.1 ± 6.1	26.9 ± 5.1	−0.2 ± 4.4	23.9 ± 5.6	24.0 ± 6.0	23.2 ± 5.3	0.1 ± 2.0	−0.7 ± 2.9	−0.6 ± 3.1
SN/GoGn(°)	36.4 ± 7.6	35.7 ± 7.7	−0.8 ± 3.1	33.6 ± 6.0	33.6 ± 5.0	32.5 ± 5.0	−0.1 ± 2.8	−1.1 ± 3.2	−1.0 ± 4.1
Occlusal plane (°)	18.1 ± 6.2	14.7 ± 7.6	−3.3 ± 4.1	14.1 ± 5.1	12.6 ± 4.5	12.5 ± 4.6	−1.5 ± 2.6	−0.1 ± 3.2	−1.6 ± 3.0
Gonial angle (°)	133.2 ± 5.9	133.1 ± 7.3	−0.1 ± 3.7	130.0 ± 6.9	129.0 ± 6.5	126.6 ± 7.0 @	−1.0 ± 4.0	−2.4 ± 2.5	−3.4 ± 5.4
UI to Pal (°)	96.3 ± 6.9	107.9 ± 8.0 #	11.6 ± 8.9	108.1 ± 11.0 @	111.8 ± 7.5	114.5 ± 5.7	3.7 ± 8.6	2.3 ± 7.3	6.1 ± 11.2 @
UI to NA (°)	11.4 ± 8.3	23.9 ± 7.2 #	12.5 ± 7.5	19.7 ± 11.7	23.0 ± 8.7	26.8 ± 6.8	3.2 ± 7.7	3.9 ± 7.9	7.1 ± 9.9 @
LI to GoGn (°)	87.6 ± 7.0	86.6 ± 6.3	−1.0 ± 6.7	91.4 ± 8.3	89.5 ± 6.7	90.6 ± 7.3	−1.8 ± 3.8	−0.8 ± 5.3	−1.1 ± 4.2
LI to NB (°)	19.2 ± 7.6	20.2 ± 7.5	1.0 ± 7.0	22.6 ± 5.8	21.6 ± 6.3	21.5 ± 6.2	−1.0 ± 3.7	−0.1 ± 5.1	−1.1 ± 4.2
InterI angle (°)	149.0 ± 9.6	138.5 ± 12.2 #	−10.5 ± 10.6	136.2 ± 11.4 @	133.3 ± 9.6	131.6 ± 6.3	−2.8 ± 10.2	−4.7 ± 13.0	−7.5 ± 13.8
Overjet (mm)	−4.4 ± 3.1	−2.6 ± 4.1	1.8 ± 3.7	−1.3 ± 2.7	1.0 ± 3.2	2.2 ± 2.8 *@	2.3 ± 3.3	1.2 ± 2.4	3.5 ± 3.3
Overbite (mm)	3.1 ± 3.0	2.1 ± 2.8	−1.0 ± 2.7	1.3 ± 2.0	1.4 ± 1.5	1.2 ± 1.9	0.1 ± 1.6	−0.2 ± 1.7	−0.1 ± 2.4

T0 (baseline), T1 (1.5-years) and T2 (3.5-years). A difference is considered significant if *p* < 0.05. * indicates a significant difference in BAMP group between T0 and T2, or between ΔT1-T0 and ΔT2-T1; # indicates a significant difference in control group between T0 and T2; @ indicates a significant difference between control and BAMP groups at T0, T2 or ΔT2-T0.

## Data Availability

The data presented in this study are available on request from the corresponding author. The data are not publicly available due to privacy reasons.

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
