# Peer review of "Skeletal Changes in Growing Cleft Patients with Class III Malocclusion Treated with Bone Anchored Maxillary Protraction—A 3.5-Year Follow-Up"

_jcm, 2021, doi:10.3390/jcm10040750_

Round 1

Reviewer 1 Report

Dear authors thank you for submitting your article.

The aim of the present study is to to investigate the 3D skeletal effect after 3.5-years of BAMP treatment in growing cleft patients using CBCT scans.

-The Introduction does not provide sufficient background, need to be extended

-Please provide the data used for the power calculation.

-Please move the paragraph 2.2(Trial Registration and Ethical Approval)

before  the paragraph 2.1 (Treatment and Control subjects)

-Please provide images with better quality

-you should include this paragraph in the materials and method ( in the last part of the Treatment and Control subjects) and not in the results "Out of the 23 initially included cleft lip and palate patients four had to be excluded for data analysis due to CBCT acquisition errors. The age of the 19 included study subjects was 11.4 ± 0.7-years at T0. For the control group, a total of 17 patients met the inclusion criteria and were included with a mean age of 10.7 ± 1.2years at T0. With regard to the protocol of elastics wearing, a high degree of compliance was recorded for all subjects based on self-report. In 7 patients a removable bite plate was used temporarily to remove the occlusal interference."

-Can you specify in the text how long did the bollard treatment last?

-if the treatment with the bollards lasted 3.5 years, did you not think that there could be TMJ problems?

-DISCUSSION

"Here, with a follow-up of 3.5-years the difference between the BAMP and the control group is most notable at the skeletal level as measured at the ANB angle, with an average increase of 3.3° that is not only statistically significant, but highly clinically meaningful."  I can't find this result in the Table, could you explain this sentence.

Reviewer 2 Report

The author says "Skeletal Changes in Growing Cleft Patients with
We commend you for your difficult research on "Class III Malocclusion Treated with BAMP".
The overall study method was good. Pictures and diagrams were also appropriate.
However, there is some misunderstanding in the statistical results of the core content of the paper, that is, the 2D comparison with the control group.
That is, the growth of the maxilla in the BAMP group shown in 3D is not clearly shown in the comparative statistics of the two groups. For example, SNA has no statistical difference. ANB has a statistical difference, but the difference is 1.1, which makes it difficult to rule out measurement errors.
In addition, in the 3D study of the BAMP group, spontaneous growth across the face seems to have occurred proactively in 3 years. In this case, there will be enough vias in the interpretation of the results, which needs to be supplemented.

Round 2

Reviewer 1 Report

Now the article has been improved. The paper is suitable for publication.

Please consider adding as a reference this article in the introduction section: 

https://doi.org/10.3390/jcm9041159

Author Response

Dear reviewer,

Thank you for your critical review of our manuscript. we are very pleased to hear that the revised munuscript is suitable for publication. We have considered your suggestion to add the reference (https://doi.org/10.3390/jcm9041159) to our introduction. However, we could not find an appropriate way to proces this reference in our manuscript. We hope our manuscript is still acceptable for publication.

Kind regards,

On behalf of all authors, Ralph Steegman